# KNOWLEDGE INFUSED DECODING

**Ruibo Liu**♣, **Guoqing Zheng**♠, **Shashank Gupta**♠, **Radhika Gaonkar**♠,
**Chongyang Gao**♦, **Soroush Vosoughi**♣, **Milad Shokouhi**♠, **Ahmed Hassan Awadallah**♠

♣Dartmouth College, ♠Microsoft, ♦Northwestern University

♣{ruibo.liu.gr, soroush.vosoughi}@dartmouth.edu

♠{zheng, shagup, ragaonka, milads, hassanam}@microsoft.com

♦cygao@u.northwestern.edu

## ABSTRACT

Pre-trained language models (LMs) have been shown to memorize a substantial amount of knowledge from the pre-training corpora; however, they are still limited in recalling factually correct knowledge given a certain context. Hence, they tend to suffer from counterfactual or hallucinatory generation when used in knowledge-intensive natural language generation (NLG) tasks. Recent remedies to this problem focus on modifying either the pre-training or task fine-tuning objectives to incorporate knowledge, which normally require additional costly training or architecture modification of LMs for practical applications.

We present **K**nowledge **I**nfused **D**ecoding (KID)—a novel decoding algorithm for generative LMs, which dynamically infuses external knowledge into each step of the LM decoding. Specifically, we maintain a local knowledge memory based on the current context, interacting with a dynamically created external knowledge trie, and continuously update the local memory as a knowledge-aware constraint to guide decoding via reinforcement learning. On six diverse knowledge-intensive NLG tasks, task-agnostic LMs (e.g., GPT-2 and BART) armed with KID outperform many task-optimized state-of-the-art models, and show particularly strong performance in few-shot scenarios over seven related knowledge-infusion techniques. Human evaluation confirms KID's ability to generate more relevant and factual language for the input context when compared with multiple baselines. Finally, KID also alleviates exposure bias and provides stable generation quality when generating longer sequences. Code for KID is available at https://github.com/microsoft/KID.

## 1 INTRODUCTION

Pre-trained language models (LMs) have been shown to capture rich semantic and syntactic features, as demonstrated by the state-of-the-art performance on many generation tasks such as abstractive summarization (Zhang et al., 2020b; Chu & Liu, 2019) and dialogue generation (Roller et al., 2021; Zhang et al., 2020c). Their generations, however, can be quite limited in scenarios requiring *knowledge*—they frequently struggle with issues such as being easily misguided by phrase co-occurrence (e.g., *"Talk? Birds can talk."*), failing to handle negation (e.g., *"The theory of relativity was **not** developed by Einstein."*) (Kassner & Schütze, 2020), and being unable to compare common-sense concepts, such as time (Qin et al., 2021) and digits (Talmor et al., 2020).

To enhance the performance of LMs on knowledge-intensive NLG tasks[1], prior studies have proposed to re-train LMs with knowledge-aware objectives (Zhou et al., 2021; Xiong et al., 2020; Zhang et al., 2019; Khandelwal et al., 2020) or add special architectures to encode knowledge (Bosselut et al., 2019; Logan et al., 2019; Peters et al., 2019b) from external resources (e.g., knowledge graphs such as CONCEPTNET (Speer et al., 2017) and ATOMIC (Sap et al., 2019)). These methods, though yielding impressive results on many downstream tasks, can be computationally expensive. More importantly,

---

[1]We define knowledge-intensive NLG tasks as those whose input context alone does not provide complete knowledge for a legitimate and plausible generation.

knowledge *implicitly* parameterized in LM architectures is difficult to revise and expand (Lewis et al., 2020b), and wrong generations are hard to diagnose due to lack of interpretation (Talmor et al., 2020), which heavily limits their real-world applications.

More recent retrieval-based models try to tackle these problems by augmenting inputs with retrieved knowledge evidence (Lee et al., 2019; Guu et al., 2020). For example, RAG (Lewis et al., 2020b) leverages non-parametric memory to access extensive knowledge (in the form of unstructured documents), and jointly fine-tunes a parametric LM (i.e., BART (Lewis et al., 2020a)) to enable knowledge-aware generation. A key limitation of such methods is that they retrieve documents only once while grounding them in the input *static* context, and thus cannot support the *dynamic* nature of the context as new tokens are generated. The static knowledge becomes a major problem for tasks where longer and abstractive generation is expected, such as open-ended story generation (Mostafazadeh et al., 2016), multi-turn dialogues (Zhao et al., 2020), and conversation summarization (Gliwa et al., 2019). Moreover, in a recent study, Krishna et al. (2021) replaced the knowledge retriever in RAG with a random retriever and found little difference in the resulting performance on a long-form QA task named ELI5 (Fan et al., 2019b), indicating the model may not be actually grounding its text generation to the retrieved documents.

To address these limitations, in this work, we present a novel decoding algorithm KID, aiming to better infuse knowledge into generation in a dynamic manner. Instead of solely relying on the *static* knowledge retrieved at beginning, during each step of LM decoding, KID *dynamically* searches promising continuation from retrieved knowledge, to guide the current step generation. Specifically, KID maintains a local knowledge memory, interacts it with a knowledge trie dynamically created from retrieved supporting documents, and updates the local memory as a knowledge-aware constraint to guide the generation. The key intuition behind KID is that existing LM pre-training objectives are usually defined at the token level yet do not explicitly model concept-centric knowledge (Xiong et al., 2020) — thus motivating us to reshape the probability mass at each step decoding towards the distribution of entities in knowledge.

The contribution of this work is three-fold: *First*, we introduce KID as a model and task agnostic decoding method that integrates knowledge on the fly and can be applied to various knowledge-intensive tasks with different generative LMs. *Second*, from a docoding perspective, on six knowledge-intensive NLG tasks, GPT2 (Radford et al., 2019) and BART (Lewis et al., 2020a) equipped with KID significantly outperform conventional beam search or sampling decoding by a large margin. *Third*, from a knowledge infusion perspective, unlike seven strong knowledge-infusion baselines which require either additional retraining or special architecture modifications, KID leverages knowledge more effectively as a light-weight knowledge infusion solution. Additionally, in few-shot scenarios KID significantly improves over them, demonstrating its generalization ability in low-resource and domain shifting regimes.

## 2 RELATED WORK

We briefly review existing work enhancing LMs with external knowledge and representative decoding algorithms for generation.

**Enhancing Language Model with Knowledge.** Large language models implicitly encode knowledge in their parameters but with limits (Petroni et al., 2019; Kassner & Schütze, 2020; Lin et al., 2020a). Several architectures and objective functions have been proposed to explicitly encode external knowledge (Sun et al., 2020; Logan et al., 2019; Roberts et al., 2020; Levine et al., 2020), or to augment LM pre-training data with retrieved knowledge (Lewis et al., 2020b; Guu et al., 2020; Lee et al., 2019). However, Talmor et al. (2020) notes that the reasoning ability of such LMs is strongly tied to the context seen during pre-training and is thus hard to generalize to new domains. Built on LMs, KID incorporates extra knowledge from external resources (Wikipedia) and thus shows strong performance in knowledge-intensive NLG tasks.

**Better Decoding Algorithm.** Two common strategies dominate the decoding algorithms used by most generative models: beam search which maximizes likelihood in a local horizon (due to finite beam size), and sampling decoding (e.g., top-$k$ sampling (Fan et al., 2018; Holtzman et al., 2018)) which relies on randomness. Holtzman et al. (2020) find beam search often produces generic and repetitive generation, while top-$k$ sampling tends to pick unlikely tokens which creates incoherent and

unrelated sequences. Existing attempts to mitigate these problems include reranking (Adiwardana et al., 2020; Shao et al., 2017), adding control signals (Zhang et al., 2018; Xing et al., 2017), and self-adaptive truncation (Welleck et al., 2020; Peters et al., 2019a). None of these decoding algorithms consider integrating knowledge in the generation process. Reflective decoding (West et al., 2021) and DeLorean (Qin et al., 2020) are two recent decoding algorithms that focus on abductive commonsense reasoning. Reflective decoding in particular has the potential to be extended to other knowledge-intensive tasks. We compare it with KID in our experiments.

## 3    KNOWLEDGE INFUSED DECODING

We detail the implementation of KID in this section. As shown in Figure 1, KID comprises of three steps: retrieving relevant knowledge (§3.1), constructing external and local knowledge memory (§3.2), and guiding current step decoding under the constraint of the knowledge trie (§3.3).

### 3.1    RETRIEVING EXTENSIVE KNOWLEDGE FROM WIKIPEDIA

The first step of KID is to retrieve several context-relevant documents to ground the following generation. We use DPR (Karpukhin et al., 2020) as our general-purpose knowledge retriever, which projects contexts and relevant documents to a 768-dim shared embedding space using a bi-encoder network (i.e., two independent BERTs (Devlin et al., 2019)). Here, the documents refer to the 100-word chunks of Wikipedia passages released by RAG (Lewis et al., 2020b), a total of 21M documents as our knowledge source $\mathbb{Z}$. We pre-load the weights from the latest checkpoint of DPR (March 2021), as it improves retrieval performance by using mined negative samples and contrastive learning, which is also suggested by Jernite (2020). During retrieval, we perform a maximum inner-product search (MIPS) with faiss[2] accelerated by GPU (Johnson et al., 2019). Formally, we retrieve $k$ most relevant document $z_{[1,...,k]} \in \mathbb{Z}$ for context $x_{\text{context}}$ as:

$$z_{[1,...,k]} = \left\{ z_i \in \mathbb{Z} \,|\, \text{topk}\left\{ \text{BERT}(x_{\text{context}})^\top \cdot \text{BERT}(z_i) \right\} \right\} \tag{1}$$

where $\text{BERT}(\cdot)$ means the vectors encoded by BERT. The number of retrieved documents $k$ is a task-specific hyper-parameter—we discuss its impact on performance in §4.3.

### 3.2    CONSTRUCTING EXTERNAL AND LOCAL KNOWLEDGE MEMORIES

We convert multi-document input retrieved from previous step into compressed knowledge memories, in order to 1) allow relevant knowledge to be easily identified, 2) reduce the memory footprint of the knowledge, whose size grows linearly with the number of retrieved documents.

Following design choice of previous successful methods (Bosselut et al., 2021; Huang et al., 2020; Fan et al., 2019a), we adopt co-reference resolution and open information extraction (OpenIE) (Stanovsky et al., 2018) to convert plain text into triplet form[3]. For example, knowledge statement like *"Iceland is a Nordic island country in the North Atlantic Ocean and it is the most sparsely populated country in Europe."* will be converted to **subject-relation-object** triplets such as ⟨***subj***:*Iceland*, ***rel***:*is*, ***obj***:*Nordic island country*⟩, ⟨***subj***:*Iceland*, ***rel***:*is*, ***obj***:*most sparsely populated country in Europe*⟩, etc. To account for overlapping elements from these triplets, we use a prefix tree (namely Knowledge Trie $G_{\text{ext}}$) to store and organize the extracted triplets, by setting the subject in each triplet as the key in the $G_{\text{ext}}$. Note that unlike common character-level dictionary tries (Chen et al., 2020; Germann et al., 2009), in $G_{\text{ext}}$ each triplet is stored in token unit as our goal is to efficiently query and traverse the knowledge triplets stored in it.

A tree structure encoding knowledge is appealing to knowledge intensive NLG tasks, since 1) the *non-cyclic* structure helps reduce repetitions in generations, and 2) querying a prefix tree can be efficiently completed in constant time ($O(|x_{\text{context}}|)$) which does not involve any costly traversal on the graph (Ji et al., 2020; Zhang et al., 2020a), regardless of the scale of grounding knowledge (normally $|x_{\text{context}}| \ll |k$ Wiki docs$|$). We also maintain a local memory $G_{\text{loc}}$ (a first-in-first-out list)

---

[2] The faiss project can be found here: https://github.com/facebookresearch/faiss.

[3] As an empirical trick we also remove the stop words in the documents as they seldom carry knowledge.

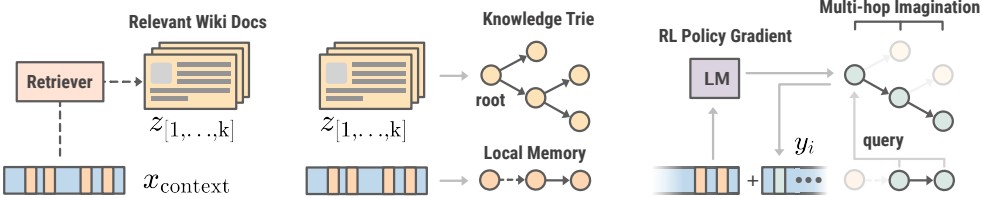

Figure 1: Overview of our KID decoding algorithm. For a given context $x_{\text{context}}$, we first retrieve $k$ most relevant Wikipedia documents $z_{[1,...,k]}$ with a knowledge retriever (Step 1), and then convert them into compressed knowledge trie $G_{\text{ext}}$ (Step 2). Meanwhile, the local memory $G_{\text{loc}}$ which is a first-in-first-out list will keep track of entities in current context, and in the final step (Step 3), it will continuously query the knowledge trie with max number of hops $h_{\text{max}}$. Current step LM decoding will be guided by the query results with policy gradient, to generate new token $y_i$.

that records all the mentioned entities in current context, to focus on concept-centric knowledge (Zhou et al., 2021; Lin et al., 2020b). More details about how we construct and query these knowledge memories can be found in §A.2 of Appendix.

## 3.3 KNOWLEDGE TRIE CONSTRAINED DECODING VIA POLICY GRADIENT

**Background.** Current generative LMs are trained to maximize the probability of generating ground-truth tokens at each decoding step. Assuming $\boldsymbol{y}_{1:T}^* = \{y_1^*, y_2^*, ..., y_T^*\}$ is the ground-truth output sequence for a given context $x_{\text{context}}$, the MLE objective minimizes the following loss:

$$J_{\text{MLE}} = -\sum_{i=1}^{T} \log p(y_t^* | y_1^*, ..., y_{t-1}^*, x_{\text{context}}) . \tag{2}$$

In knowledge-intensive NLG tasks, however, it is reported that the MLE training goal does not explicitly model knowledge, and thus the LM often produces counterfactual generation by surface-level misguidance (Zhou et al., 2021; Petroni et al., 2019). Furthermore, the teacher-forcing algorithm used by MLE training leads to the exposure bias problem (Ranzato et al., 2016), as the LM has access to ground truth sequence up to the next token during training, but does not have such signal during testing, which causes accumulated errors when generating longer sequence (Paulus et al., 2018). Both problems heavily limit the performance of popular LMs on diverse knowledge-intensive NLG tasks. One remedy is to learn a generation *policy* that not only maximizes knowledge correctness but also alleviates exposure bias in longer generation, which can be made possible with policy gradient in reinforcement learning (RL).

**Knowledge Trie Constrained Decoding.** To formulate NLG as a RL problem, we define the *state* as the generated tokens before $t$ (i.e., $s_t = y_{<t}$), and the *action* as the current step output token (i.e., $a_t = y_t$). The softmax output of the language modeling head, i.e., a categorical distribution $p_t$ over the entire vocabulary, is considered as the policy $\pi_t$ for picking token $y_t$ (action $a_t$) given the state $s_t = y_{<t}$ (Guo et al., 2021; Liu et al., 2020). Note that $p_t$ (i.e., $\pi_t$) could be from either a pre-trained LM or a LM fine-tuned with task data. While conventional sampling decoding and beam search pick next token directly from $p_t$, here we look to inject knowledge to adjust $p_t$ to guide decoding.

For current input context $x_{\text{context}}$ at step $t$, we first take its concept mentions in local memory $\{c_1, c_2, ..., c_m\} \subset G_{\text{loc}}$, and then query the knowledge trie $G_{\text{ext}}$ with these concepts by max hops of $h_{\text{max}}$. The union of queried results of all hops $\mathbb{V}_i = \{v_1^i, v_2^i, ..., v_n^i\}$ for $i = 1, .., h_{\text{max}}$ then serve as knowledge demonstrations for current step generation. To put probability mass on tokens aligning with these demonstrations, we compute the $t$-th step knowledge gain $r_t$ as the total log probability of all retrieved tokens from $G_{\text{ext}}$ under decoding policy $\pi_t$ as $r_t = \sum_{i=1}^{h_{\text{max}}} \sum_{v \in \mathbb{V}_i} \mathbb{I}[v]^\top \cdot \log \pi_t$, where $\mathbb{I}[\cdot]$ is the indicator function that will output an one-hot vector with a 1 in the coordinate of token $v$

in the vocabulary and 0's elsewhere. With $r_t$, we define the $t$-th step reward $J_{\text{RL},t}$ on trajectories $\tau$ induced by $\pi_t$ as:

$$J_{\text{RL},t} = \mathbb{E}_{\tau \sim \pi_t} \left( \frac{\pi_t^*(a_t|s_t)}{\pi_t(a_t|s_t)} \cdot r_t \right) - \beta \text{KL}(\pi_t || \pi_t^*), \tag{3}$$

where $\pi_t^*$ is the desired policy (a vector initialized from $\pi_t$) to produce tokens approaching knowledge demonstrations, and the KL penalty term $\text{KL}(\pi_t || \pi_t^*)$ is to avoid the updated policy drifts too much away from the original one, also known as *trust region* constraint (Schulman et al., 2017; 2015). Note that we use off-policy sampling to collect trajectories, with an importance weight $\pi_t^*/\pi_t$ to calibrate the knowledge gain $r_t$, which can stabilize the optimization (Munos et al., 2016).

---

**Algorithm 1:** Trie-Constrained Policy Gradient

**for** $t = 1, 2, \ldots$ **do**
    Collect samples $(a_t|s_t)$ by vanilla policy $\pi_t$;
    Compute reward $J_{\text{RL},t}$ by Eq. (3);
    Compute updated policy
    $\pi_t^* \leftarrow \arg\max_{\pi_t^*} J_{\text{RL},t}$ by taking $K$ steps of
    SGD (via Adam);
    **if** $\text{KL}(\pi_t || \pi_t^*) \geq 2\sigma$ **then**
        | $\beta_{t+1} = 2\beta_t$;
    **else if** $\text{KL}(\pi_t || \pi_t^*) \leq \sigma/2$ **then**
        | $\beta_{t+1} = \beta_t/2$;
    **end**
    Generate token $y_t$ with updated policy $\pi_t^*$;
**end**

Algorithm 1 shows how we obtain the updated policy through policy gradient. We set $\beta$ dynamically to control the KL penalty within the reward function. The target divergence $\sigma$ tunes the strength of knowledge infusion—smaller $\sigma$ means less infusion while larger $\sigma$ provides more space for policy gradient (We set $\sigma$ to 0.02 across all tasks). To generate the actual token for step $t$, we pick from the updated policy $\pi_t^*$ with sampling decoding. The token should conform to the knowledge demonstrations, since its corresponding hidden states have shifted towards $\mathbb{V}$ due to policy gradient. We empirically choose $K = 3$ for good performance in most cases.

Prior approaches have also explored to set sequence-level metrics such as BLEU (Papineni et al., 2002) as the reward to directly optimize the generation quality (Li et al., 2016; Paulus et al., 2018). However, many studies report such sparse reward will cause low efficiency optimization (i.e., $J_t = 0$, $\forall t < T$) (Guo et al., 2021). Our trie-constrained policy gradient method seemingly mitigates this problem by using an immediate reward ($J_{\text{RL}}^{\theta^*}$) at each step with reasonable approximation. Recent off-line RL for NLG work show promising results when using data itself (rather than metrics) to estimate rewards (Pang & He, 2021; Jaques et al., 2020)—our design of knowledge trie echoes their findings. The gold data distribution memorized in trie is treated as guidance to reshape each step probability distribution, and thus brings benefits on alleviating exposure bias during long generation.

## 4 EXPERIMENTS

### 4.1 EXPERIMENTS SETTINGS

We consider three diverse types of knowledge-intensive tasks for evaluation (statistics see §A.1):

**Abstractive Question Answering.** We study Abstractive QA, which requires the model to *generate* free-form answers to the questions. We choose long-form QA task ELI5 (Fan et al., 2019b) and MSMARCO NLG task v2.1 (Nguyen et al., 2016) as two commonsense QA tasks whose questions can be mostly answered by referring to Wikipedia passages. We also use two extra QA tasks PIQA (Bisk et al., 2020) and PubMedQA (Jin et al., 2019) whose questions require domain-specific knowledge to answer (i.e., physical interaction knowledge for PIQA, and medical knowledge for PubMedQA). We calculate BLEU-1 and ROUGE-L scores to be able to compare directly with related work (Lewis et al., 2020b; Krishna et al., 2021).

**Logic-centric Writing.** We also investigate whether KID can benefit NLG tasks that do not have an explicit query form for certain knowledge (i.e., with specific questions, like QA tasks). We study ROC story ending generation (Mostafazadeh et al., 2016), which requires generating a legitimate ending given a four-sentence context, and $\alpha$NLG (Bhagavatula et al., 2020) which

Table 1: Benchmark results on six diverse knowledge-intensive tasks. Compared with beam search (Beam) and sampling decoding (Sample), KID decoding improves the generation quality in general (by at most $15\%$). For each LM, we report their performance in zero-shot setting (*), and that of being fine-tuned (FT). We color (▭ ▭) those results of KID that achieve > $5\%$ improvement over the next-best performance. We also annotate the performance reported by published state-of-the-art models to date (- means missing official reports).

| | ELI5 | | MSMARCO | | ROC | | $\alpha$NLG | | WoW | | MuTual | |
|---|---|---|---|---|---|---|---|---|---|---|---|---|
| **Existing Method** | B-1 | R-L | B-1 | R-L | B-1 | R-L | B-1 | R-L | F-1 | R-L | $MRR_G$ | R-L |
| **GPT2-M** [345M]* | 14.6 | 16.1 | 28.8 | 30.1 | 12.0 | 13.9 | 13.2 | 15.0 | 9.3 | 10.8 | 29.7 | 7.1 |
| - FT + Beam | 22.9 | 24.6 | 44.1 | 50.6 | 26.4 | 20.6 | 19.4 | 25.7 | 10.3 | 12.1 | 47.9 | 11.3 |
| - FT + Sampling | 23.8 | 25.4 | 46.2 | 51.2 | 26.0 | 20.0 | 18.9 | 25.1 | 12.6 | 11.9 | 51.6 | 20.3 |
| - FT + KID | **27.9** | **26.6** | **47.4** | **53.9** | **28.1** | **21.2** | **21.7** | **26.9** | **16.4** | **15.9** | **53.3** | **22.4** |
| **BART-L** [406M]* | 21.4 | 20.6 | 19.1 | 19.7 | 12.5 | 16.7 | 24.6 | 28.0 | 8.1 | 8.5 | 35.1 | 12.9 |
| - FT + Beam | 25.6 | 24.7 | 44.5 | 50.5 | 22.3 | 20.2 | 31.9 | 34.1 | 10.9 | 11.9 | 49.2 | 20.6 |
| - FT + Sampling | 25.8 | 25.1 | 48.4 | 53.5 | 24.4 | 20.9 | 30.5 | 33.6 | 12.2 | 15.0 | 53.7 | 20.4 |
| - FT + KID | **27.4** | **26.3** | **51.9** | **56.9** | **26.5** | **21.3** | **33.4** | **35.6** | **15.7** | **16.6** | **54.5** | **22.7** |
| **Published SotA$^\diamond$** | - | 26.2 | - | 57.2 | 26.3 | 20.8 | - | 45.0 | 13.5 | 15.5 | - | - |

requires generating reasonable hypothesis given two observations. We follow related work (Zhou et al., 2021; Guan et al., 2020) in using BLEU-1 and ROUGE-L as evaluation metrics.

**Dialogue Generation.** Chitchat dialogues are normally multi-turn discussions over a variety of topics or concepts, which often involve topical and factual knowledge (Petroni et al., 2021). We study two dialogue datasets that require knowledge grounding: Wizard of Wikipedia (WoW) (Dinan et al., 2019), where the speaker in the conversation must ground their utterances in Wikipedia passages, and MuTual (Cui et al., 2020), where utterances have to be a logically-coherent continuation of the given multi-turn context. We follow existing work and the leaderboard of WoW in using F-1/ROUGE-L for WoW and MRR/ROUGE-L for MuTual evaluations.

We take two representative language models to demonstrate the effectiveness of KID: 1) GPT2-medium (Radford et al., 2019) which is an auto-regressive LM, and 2) BART-large (Lewis et al., 2020a) which is a text-to-text LM. We tune the hyperparameters based on the models' performance on an in-house split dev set, and report the results that were best on the official dev set[4].

## 4.2 MAIN RESULTS ON DECODING PERFORMANCE

**Comparison with off-the-shelf decoding methods.** Table 1 compares the results of GPT2-medium and BART-large on six diverse NLG tasks with beam search (Beam), sampling (Sampling), and our proposed KID decoding algorithm. Compared with the other two commonly used decoding algorithms, knowledge-guided KID achieves better results for all the cases, with significant improvements (with p-value $p < 0.01$) over the next-best decoding strategy (e.g., 4.1 absolute increase for BLEU-1 in ELI5 (GPT2)). We also notice that KID brings greater improvements to auto-regressive language models – above 5% improvement in 9 out of 12 metrics for GPT2-medium, in contrast to 5 out of 12 for text2text language model (BART-large). The reason could be that the reinforcement learning objective of KID is more similar to the MLE objective of GPT2 than the denoising objective of BART (Pang & He, 2021; Guo et al., 2021). Compared with task-specific state-of-the-art models[5], task-agnostic LMs armed with KID can beat SOTA results on three different tasks (ELI5, ROC, WoW), which is not possible when beam search and sampling decoding is used. This interesting observation demonstrates a huge potential in inference-time optimization, which we believe is worth exploring further.

---

[4]For sampling decoding, we run experiments with all combinations of top-$p$ ($p \in [0, 0.1, ..., 1]$) and top-$k$ ($k \in [0, 10, ..., 100]$), while for beam search, we sweep the number of beams from 1 to 10. With the updated decoding policy, KID uses sampling decoding (with similar search to get optimum $p$ and $k$) to pick actual tokens.

[5]Published SotA models to date (October 2021): ELI5 (RAG; 2020b), MSMARCO (RAG; 2020b), ROC(Knowledge-enhanced GPT2; 2020), $\alpha$NLG (T5; 2019), WoW (BART+DPR; 2021), and MuTual (Human Performance; 2020). The information mainly comes from corresponding leaderboards.

Table 2: Performance of six related works on Wiki-answerable ELI5 and MSMARCO, and out-of-domain PIQA and PubMedQA QA tasks. We report ROUGE-L score with 10% and 100% of the training data. As a model-agnostic method, KID shows particularly strong performance in few-shot scenarios, which can better help LMs transfer to new domain with minimum training data.

| Method / Available FT Data | ELI5 | | MSMARCO | | PIQA | | PubMedQA | |
|---|---|---|---|---|---|---|---|---|
| | 10% | 100% | 10% | 100% | 10% | 100% | 10% | 100% |
| **GPT2 + Knowledge** (KG Triplets Post-train; 2020) | 9.3 | 15.7 | 22.3 | 42.1 | _7.5_ | 16.2 | 3.6 | 8.4 |
| **GPT2 + COMeT Emb.** (KG Embedding Fuse; 2020) | _13.4_ | 17.3 | _30.3_ | 44.6 | 6.4 | 16.5 | 4.5 | 5.8 |
| **RAG** (Wiki Retrieval Augmented; 2020b) | 5.7 | _21.4_ | 25.4 | **57.2** | 3.2 | 17.3 | 1.2 | 6.7 |
| **FiD-T5** (Two-step Retrieval + Seq2Seq; 2021b) | 3.9 | 18.1 | 23.7 | 53.1 | 4.5 | 17.4 | 1.3 | 4.5 |
| **QA-GNN** (GNN + Attention; 2021) | 6.2 | 19.5 | 21.3 | 50.5 | 6.3 | **19.1** | _7.8_ | **11.7** |
| **ReFlective** (Forward + Backward LMs; 2021) | 8.7 | 18.2 | 23.5 | 44.7 | 5.9 | 16.7 | 4.1 | 9.2 |
| **Ours:** KID (with GPT2-medium) | **15.2** | **26.6** | **32.3** | _53.9_ | **10.5** | _18.4_ | **9.9** | _11.5_ |

**Comparison with existing knowledge-infusion methods.** Besides evaluating KID from a pure decoding perspective, we also compare KID with existing methods of integrating knowledge for knowledge-intensive NLG tasks. In addition to ELI5 and MSMARCO, we also evaluate on two extra QA tasks: PIQA and PubMedQA (discussed in §4.1), where answers are considered to be *not* fully covered by Wiki knowledge. Besides **RAG** (Lewis et al., 2020b), we consider several competitive prior methods incorporating knowledge including a) **GPT2 + Knowledge** (Guan et al., 2020), which post-trains GPT2 on triplets-converted augmentation data (e.g., ⟨*helium, is, gas*⟩ → *"helium is gas."*); b) **GPT2 + COMeT Embeddings**, which fuses knowledge-aware embeddings (e.g., from CoMET (Bosselut et al., 2019); c) **FiD-T5** (Izacard & Grave, 2021b;a), which concatenates retrieved grounding documents with context as new input; d) **QA-GNN** (Yasunaga et al., 2021), which traverses a graph neural network; e) **Reflective** decoding (West et al., 2021), which relies on forward and backward LMs to encode bi-directional context for generation.

As shown in Table 2 (columns with 100% training data), KID outperforms all other methods in ELI5 (with 5.2 ROUGE-L points improvements over the second-best method (RAG)), and achieves competitive results requiring neither specific model architecture nor additional training to infuse knowledge. We also evaluate on a few-shot setting, where only 10% of task training data is available to mimic domains for which ample training data is unavailable or difficult to acquire in practice. This setting also tests a method's ability to transfer to new domain and to generalize to unseen entities, concepts or events. Also shown in Table 2, KID with a LM similar in size to the baselines (GPT2-medium) achieves best few-shot performance in all four tasks, including PIQA and PubMedQA. Our experiments find baseline methods tend to generate off-topic and hallucinatory answers when the expected answer length is long (e.g., ELI5 and PIQA). RAG shows limited performance in few-shot scenarios, due to its static knowledge retrieved by initial context cannot be generalized to newly generated tokens. KID, instead, dynamically searches references for grounding, thus showing more agility in unfamiliar context. Comparison with more baselines can be found in §A.3 of Appendix.

## 4.3 ABLATION STUDIES FOR THE BEST PERFORMING KID

**How to choose retriever?** We experiment with replacing the default DPR document retriever of KID, with popular retrievers including the TF-IDF retriever from DrQA (Chen et al., 2017), and the Wiki retriever used in BLINK (Wu et al., 2020). We also experiment with a random retriever baseline that retrieves documents randomly given the context. We choose two tasks ELI5 and WoW that provide ground-truth knowledge provenance, which are also the only two KILT tasks requiring long and abstractive generation (Petroni et al., 2021). Following KILT benchmarking metrics, we use precision@1 (Prec@1) to measure the top-1 retrieval accuracy, and ROUGE-L (R-L) to evaluate generation quality. We also consider directly measuring how much knowledge in the ground-truth evidence (provenance) appear in the actual generation. We compute knowledge Coverage (Cov) (used in Guan et al. (2020)) which is the 1-gram overlap between the triplets of provenance and the generation, to measure the extent to which our generation is actually using the retrieved knowledge.

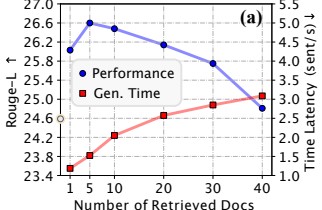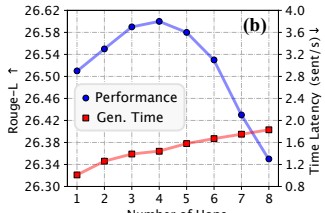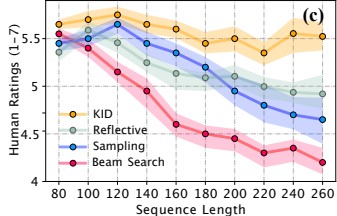

Figure 2: Impact of hyperparameters on KID's ELI5 performance when (a) more documents are retrieved, and (b) more hops taken when querying the knowledge trie. (c) Average human ratings on different-length sequences generated by KID, sampling, beam search, and reflective decoding. KID generation has more stable quality across lengths by restraining exposure bias.

Table 3 shows the corresponding results. For reference, RAG obtains nearly equivalent performance (R-L and Cov) with random retriever and the DPR[FT][6], which indicates its generation mainly relies on the fine-tuned BART and ignores retrieved knowledge (also observed in Krishna et al. (2021)). In contrast, KID with the default DPR retriever outperforms all other retrievers and RAG variants in retrieval accuracy (Prec@1), knowledge coverage (Cov) and final generation quality (R-L). We observe high correlation between (Cov, R-L) and Prec@1, indicating that a good knowledge retrieval is essential for knowledge-intensive NLG tasks.

Table 3: A closer comparison of BART-L with KID and RAG (Lewis et al., 2020b) which also leverages retrieved Wikipedia passages as knowledge. We switch between different retrievers to study its impact on retrieval accuracy (Prec@1), generation quality (R-L), and knowledge coverage (Cov).

|  | ELI5 | | | WoW | | |
|---|---|---|---|---|---|---|
|  | Prec@1 | R-L | Cov | Prec@1 | R-L | Cov |
| RAG w/. Random | 3.4 | 21.3 | 2.62 | 15.6 | 10.4 | 9.12 |
| RAG w/. DPR[FT] | 16.7 | 21.4 | 2.68 | 26.3 | 11.8 | 11.8 |
| KID w/. Random | 3.4 | 16.5 | 1.34 | 15.6 | 9.5 | 7.79 |
| KID w/. TF-IDF | 11.0 | 20.9 | 3.88 | 41.9 | 15.4 | 12.8 |
| KID w/. BLINK | 8.9 | 21.5 | 2.87 | 24.3 | 10.0 | 8.31 |
| KID w/. DPR (default) | 17.3 | 26.3 | 4.39 | 45.5 | 16.6 | 15.7 |

Table 4: The performance (R-L) on ELI5 of LMs with different sizes (similar architecture). Vanilla LMs (*) benefit more with KID than the fine-tuned ones (FT). (The absolute gain over the next-best is annotated.)

| Model | ELI5 | | |
|---|---|---|---|
|  | Beam | Sample | KID |
| GPT2-M* | 16.0 | 16.1 | 20.2 ▲4.1 |
| GPT2-M[FT] | 24.6 | 25.4 | 26.6 ▲1.2 |
| GPT3-1.3B* | 21.7 | 22.0 | 24.5 ▲2.5 |
| GPT3-1.3B[FT] | 24.9 | 25.5 | 26.6 ▲1.1 |
| GPT3-2.7B* | 22.8 | 24.6 | 26.7 ▲2.1 |

**How much knowledge do we need?** We also study the impact of *number of documents* ($k$) we retrieve and number of hops ($h_{max}$) we use to query the knowledge trie, two factors that determine how much knowledge we use to ground the generation. As shown in Figure 2 (a) and (b), for the example task ELI5, we find the generation performance measured by ROUGE-L does not benefit from simply more retrieved documents—an optimum $k$ is 5 through empirical observation, and similarly, $h_{max} = 4$ brings best performance. A larger $k$ might risk retrieving less relevant Wiki documents and a larger hop $h_{max}$ with deeper traverse through the knowledge trie tends to bring in off-topic knowledge. We also plot the average time consumed for generating each sentence as reference (the second $y$-axis in Figure 2 (a) and (b)), which demonstrate the best performing $k = 5$ and $h_{max} = 4$ achieve a reasonable trade-off between generation quality and decoding speed.

In Figure 2 (c), we quantify the exposure bias problem through human judgements. We first sample 200 ELI5 test set questions and generate answers of various lengths {80, 100, ..., 260} (260 is the average sequence length in training set) with beam search, sampling, reflective (West et al., 2021), and KID. We then ask humans to rate these generations with 7-point Likert scoring (Joshi et al., 2015) how likely the generated text is a natural sentence. Each generation receives at least 15 ratings. We observe that both beam search and sampling methods suffer from the exposure bias problem[7], since

---

[6]RAG fine-tunes the query encoder of DPR with BART, which differs the off-the-shelf DPR used in KID.

[7]We use beam size 5 for beam search, and top $p = 0.9$ and $k = 20$ for sampling decoding, as they yield best ROUGE-L score during automatic evaluation.

Table 5: Human assessments of generation in terms of Relevance, Factuality, and Grammaticality on a 7-point Likert scale. We run paired sample $t$-test comparing human references (Gold) with beam search (BM) with beam size 5, sampling (SP) with top $p = 0.9$ and $k = 20$, reflective (RFLC) decoding, and our KID generation. $p$ value describes the significance of difference from Gold. (* corresponds to $p$-value$< 0.05$ and ** to 0.01.)

| | | ELI5 | | | | | aNLG | | | | | WoW | | | | |
|---|---|---|---|---|---|---|---|---|---|---|---|---|---|---|---|---|
| | | Gold | BM | SP | RFLC | **KID** | Gold | BM | SP | RFLC | **KID** | Gold | BM | SP | RFLC | **KID** |
| **Relevance** | Mean | 5.14 | 4.51 | 4.97 | 4.73 | 5.07 | 5.42 | 5.22 | 5.10 | 5.27 | 5.36 | 4.62 | 4.68 | 4.44 | 4.35 | 4.57 |
| | $p$-value | - | .00** | .10 | .03* | .30 | - | .14 | .02* | .17 | .23 | - | .12 | .10 | .06 | .30 |
| **Factuality** | Mean | 4.95 | 4.37 | 4.61 | 4.23 | 4.87 | 5.35 | 5.17 | 5.19 | 5.25 | 5.30 | 4.72 | 4.20 | 4.38 | 4.41 | 4.53 |
| | $p$-value | - | .14 | .00** | .00** | .24 | - | .05 | .06 | .30 | .41 | - | .00** | .06* | .15 | .29 |
| **Fluency** | Mean | 5.66 | 5.52 | 5.54 | 5.07. | 5.50 | 5.40 | 5.23 | 5.34 | 4.97 | 5.27 | 4.53 | 4.48 | 4.40 | 4.14 | 4.33 |
| | $p$-value | - | .09 | .11 | .02* | .07 | - | .15 | .20 | .04* | .16 | - | .18 | .10 | .05 | .08 |

their ratings deteriorate as the length grows. Reflective decoding exhibits similar trend since it stills relies on MLE training. KID, instead, dynamically infuses global knowledge with LM predictions at each step and thus can mitigate exposure bias by imitating non-local demonstrations.

**Does the size of LM matter?** We run experiments with different sizes of LMs that have the similar architecture (GPT2-medium, and GPT3 with 1.3B, and 2.7B parameters[8]). Table 4 shows that overall larger LMs benefits all decoding methods with KID consistently outperforming Beam search and sampling decoding. In addition, KID brings more gain for non-finetuned LMs than fine-tuned ones, potentially because the fine-tuning LM is already fit onto the new domain thus less knowledge infusion is needed. Interestingly with KID, vanilla GPT3-2.7B outperforms its 1.3B fine-tuned counterpart, which is especially meaningful since ever-large foundation models (Bommasani et al., 2021) are expensive to fine-tune effectively on common hardware settings (Liu et al., 2021).

## 4.4 HUMAN EVALUATION

We recruited 300 MTurk participants to manually examine generated outputs of several decoding algorithm in terms of relevance, factuality, and fluency. Each participant was asked to review five sample generations without revealing their source. We used paired samples $t$-tests to examine the difference between human references and other decoding methods generation. As Table 5 shows, KID generates similar quality sequences as human references without significant differences across all three tasks, while in ELI5, beam search and reflective decoding generation are rated significantly low in both relevance and factuality, partially due to exposure bias in longer generation. There is no significant difference in grammaticality between KID and vanilla decoding methods. The exact questions we asked participants can be found in §A.5 of Appendix.

## 5 CONCLUSIONS AND FUTURE WORK

In this work, we proposed KID, a novel decoding algorithm which in each step of decoding dynamically fetches knowledge and guides token decoding via interaction between a local knowledge memory and a constructed external knowledge trie. Given its decoding nature, KID doesn't require architecture changes to existing LMs and is applicable to various generative LMs. Evaluations on eight knowledge-intensive NLG tasks demonstrated the effectiveness of KID and that its generations are aligned with human references well. In addition, KID is also found to alleviate exposure bias and maintain quality for long generations.

Future work of KID could study more efficient data structures to accelerate knowledge fetching and integration. One could also investigate combining KID with prompt based generations to further boost the performance, especially in the few-shot settings. Another direction is to further study the integration of KID with full size foundation models, like GPT-3 175B to understand KID's potential.

---

[8]We adopt the public implementation of GPT3—GPT-Neo (github.com/EleutherAI/gpt-neo).

ETHICS AND REPRODUCIBILITY STATEMENT

The goal of KID is to provide a general-purpose knowledge infusing decoding algorithm by leveraging retrieved Wikipedia documents. Still, the generation of KID can be affected by certain biases from the LM it is based on though those biases may be partially mitigated by the knowledge demonstrations (Liu et al., 2022; Rae et al., 2021). Another major ethical consideration is that KID can mimic undesirable attributes of the target knowledge source documents that could be non-contemporary and do not represent current norms and practices—and KID has no scheme to diagnose these problems (Lewis et al., 2020b). Furthermore, our experiments and analysis are done in English, and therefore we do not claim that our findings will generalize across all languages, although our framework has potential to be extended to other languages with necessary modifications.

For reproducibility, evaluations of KID and baseline methods are all conducted on public NLG data sets. We compare results from published papers and public leaderboards. Code and scripts for reproducing KID is available on GitHub at `https://github.com/microsoft/KID`.

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

# A APPENDIX

## A.1 DATASETS STATISTICS

We chose eight knowledge intensive NLG tasks to evaluate KID. Here we present the dataset statistics of these tasks in Table A1. The links to these datasets can be found in §4.1.

Table A1: The dataset statistics of the eight knowledge-intensive NLG tasks we evaluate for KID.

| Split | ELI5 | MSMARCO | PIQA | PubMedQA | ROC | $\alpha$NLG | WoW | MuTual |
|-------|------|---------|------|----------|-----|-------------|-----|--------|
| Train | 272,764 | 153,725 | 16,115 | 800 | 52,665 | 50,481 | 63,734 | 7,088 |
| Dev | 1,507 | 12,467 | 1,838 | 100 | 1,571 | 7,252 | 3,054 | 886 |
| Test | 600 | 12,467 | 3,000 | 100 | 4,081 | 14,313 | 2,944 | 886 |

## A.2 DETAILS ON KNOWLEDGE TRIE CONSTRUCTION AND QUERY

### A.2.1 KNOWLEDGE TRIE CONSTRUCTION

In this section, we detail the process of constructing the knowledge trie and how we query for the knowledge in a dynamic fashion. In Figure A1 (a), for a given question (say from the ELI5 dataset), the DPR retriever (Karpukhin et al., 2020) would retrieve $k$ documents (the effect of choosing different $k$ on performance has been discussed in §4.3; we use $k = 3$ here for simplicity) from 21M 100-token Wiki documents as the grounding passages for the question. We then use co-reference resolution to replace the pronouns with their referents (colored in red), normalize the text (e.g., removing links, lower-casing, etc.), and pass them through the OpenIE (Stanovsky et al., 2018) to obtain knowledge triplets (the use of OpenIE can be also seen in related prior work (Trisedya et al., 2019; Wu et al., 2019; Fan et al., 2019a)). The end nodes of the extracted triplets (i.e., the *subj* and *obj*) serve as the key-value pairs when they are stored in the external knowledge trie ($G_{ext}$), and the relations between nodes are translated to the edges. We use the stems of the tokens as the keys in $G_{ext}$ (e.g., "*driving*" → "*drive*"), in order to compress duplicated nodes in a concept-centric manner.

During knowledge trie construction, besides the one-step key-value pairs from the triplets, we also consider multiple-step relations by looking for key mentions in the values of previous keys (e.g., value: "*influence of drugs*" → next-hop key: "*drug*"). Such iterative procedure will end when there are no more keys appearing in the values. Specifically, we use depth-first search (DFS) to record the maximum depth of the child branch for each key node, and also memorize the corresponding values (and its next-hop key) on the branch to facilitate further query.

### A.2.2 DYNAMIC QUERYING FOR KNOWLEDGE

Since retrieving knowledge from millions of documents is slow (even with GPU acceleration), the knowledge trie described above is constructed off-line. We pre-compute and store this knowledge trie on the disk, and build the local knowledge memory on the fly. The local knowledge memory is simply a First-in-First-out (FIFO) list which continuously stores newly generated entities ($G_{loc}$; initialized with entities in the input question), whose length will be limited by $w_{max}$ (we set $w_{max} = h_{max}$ empirically, where $h_{max}$ is the max query hops). In Figure A1 (b) we show a query example with the query word "*driving*" in the local memory. The query word is firstly converted to its stem (i.e., "*drive*"), and then its demonstrations are retrieved with the key "*drive*".

We show the whole generation loop of KID in Algorithm 2. At each step of the decoding, we first collect knowledge demonstrations by querying $G_{ext}$ with each entity in $G_{loc}$ as key. The collected demonstrations (a list of tokens) will then serve as target word space to guide current step generation (has been detailed in §3.3). The local knowledge memory is updated only when a new entity is generated. This is because the non-entity words (such as stop words, digits, etc.) a) are rarely the keys in the $G_{ext}$ (which are *subj* in OpenIE triplets), and b) rarely lead to meaningful next-step constraints (it's hard to answer what's a proper continuation of "*the*").

Since the external knowledge trie is constructed off-line, the query time mainly depends on the number of query words in the local knowledge memory (with max length $w_{max}$), and the maximum number of

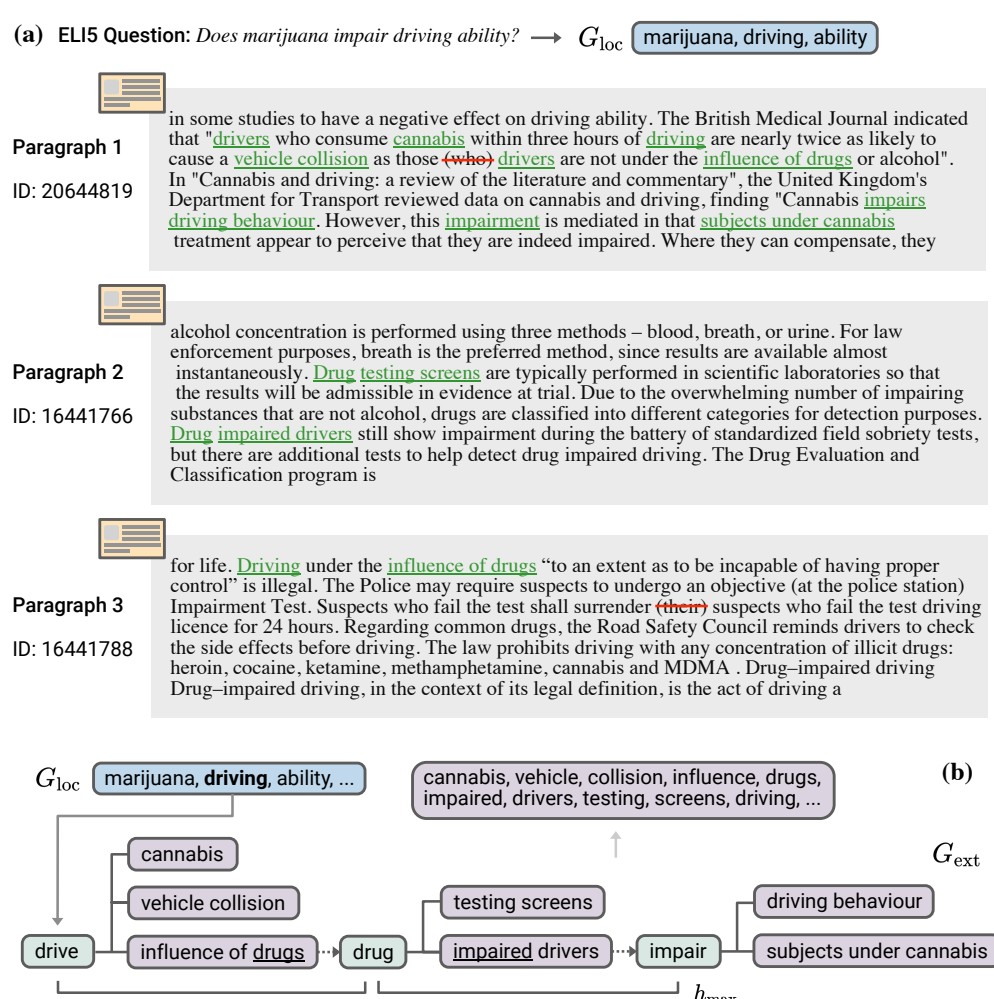

Figure A1: A sample of KID's retrieved documents for the ELI5 question "*Does marijuana impair driving ability?*" and its corresponding knowledge trie. **(a)** The top three relevant documents retrieved by DPR. We annotate the triplets end nodes (*subj* and *obj*) picked by OpenIE for knowledge trie construction in green. **(b)** The partially observed knowledge trie when we query "*driving*" in the current local knowledge memory (blue) by $w_{max}$ hops. We perform co-reference resolution to replace pronouns with actual entities in the documents, and use the stems of the tokens as the query words and keys in the external knowledge trie. The retrieved demonstrations (values in the trie; purple) in multiple hops will then serve as guidance for current step decoding, after split into single tokens.

hops we query the external knowledge trie ($h_{max}$) with, which is approximately $w_{max} * h_{max}$ in total. We claim this is actually constant time complexity because $h_{max}$ and $w_{max}$ are fixed hyper-parameters (normally 1-5, as discussed §4.3) and it will not scale up with sequence length or the number of grounding documents retrieved.

## A.3   MORE COMPARISON WITH IMPROVED DECODING ALGORITHMS

Besides the baselines we compare in the paper, there are other inference-time decoding methods that focus on improving generation quality in different perspectives, such as diversity (Baheti et al., 2018), and attributes controlling (Dathathri et al., 2020), and entity appearance (Mao et al., 2020)[9].

---

[9]We cannot find its official implementation, and thus we do not include it in the following comparison.

---

**Algorithm 2:** The Generation Loop of KID

---

**Input:** constructed knowledge trie $G_{ext}$, local knowledge memory $G_{loc}$, input context $q$, max query hop $h_{max}$, max sequence length $L$.

**Output:** generated tokens $\{x_1, x_2, ..., x_t\}$.

$G_{loc} \leftarrow$ entities in $q$;

**while** current sequence length $< L$ **do**

    Knowledge demonstrations $D \leftarrow []$;

    **for** each entity $e$ in $G_{loc}$ **do**

        Query $G_{ext}$ with $e$ by $h_{max}$ hops to collect demonstrations, and store in $D$.

    Use $D$ to guide current step decoding by Algorithm 1, and generate token $x_t$.

    **if** $x_t$ is entity **then**

        Append $x_t$ to $G_{loc}$, and slim $G_{loc}$ by window size $w_{max}$.

**return** $\{x_1, x_2, ..., x_t\}$

---

We also notice some methods that are training-time optimization, such as CTRL (Keskar et al., 2019), which requires re-training language models conditioned on a set of static control words, and fusion-in-decoder (FiD) (Izacard & Grave, 2021b;a), which concatenates additional retrieved grounding sentences with input text, and fine-tunes text2text language models (e.g., T5 (Roberts et al., 2020)). KID differs from all these methods, since 1) KID aims to improve knowledge awareness during decoding, though we also find KID could mitigate exposure bias, and 2) KID can function as a standalone module requiring neither specific model architecture nor re-training or fine-tuning the language model (as shown in §4.3, the experiments with vanilla GPT3-2.7B). In the following part, we compare KID with all above methods except CTRL since it requires costly language model retraining on each new domain and how to pick control words for dynamic knowledge seems ambiguous. We also prepare a naive version of KID (Little KID) which uses plain list instead of graph structure (the knowledge trie) to store knowledge, where we directly use all the end nodes of extracted triplets from the documents as constraints without any dynamic query. We run experiments on QA tasks as FiD was trained on, and discuss their differences on performance, deployment, and efficiency (time/space).

### A.3.1 PERFORMANCE ON KNOWLEDGE AWARENESS

In Table A2 we present the comparison results about performance on knowledge awareness. Not surprisingly, FiD is the strongest baseline among others, as it also explicitly leverages retrieved documents to ground its generation. Compared with KID, we find the performance of FiD is relatively limited when the NLG tasks requires longer and creative generation. For example, ELI5, whose average reference length is around 220 tokens, and FiD tends to generate generic and repeated sequence after about 80 tokens from our observation, potentially because though FiD fuses knowledge during training, there is no scheme to guarantee the knowledge parameterized in the model can be expressed during inference. Diverse Decoding, and PPLM show mediocre performance in these tasks because no knowledge-aware object was considered, but relatively well on longer generation tasks (e.g., ELI5), which seems to demonstrate the advantage of inference-time optimization methods (as KID). Little KID is not able to dynamically query knowledge when new context is generated, and thus performs not as good as KID, especially in longer generation (e.g., ELI5).

### A.3.2 COMPARISON OF TIME AND SPACE CONSUMPTION

The time and space cost of running KID could be a matter of concern especially when we consider real-world applications. In Table A3, we compare the time complexity of KID with other baselines, including vanilla beam search and sampling decoding. We provide a breakdown for different stages: retrieving knowledge (**retrieving**), training the generative model (**training**), and the actual generation (**inference**). Beam search and sampling decoding only have noticeable cost when there is sorting process filtering the candidate tokens (common in beam search and top-$p$ or top-$k$ sampling), which will often bring $O(k \log k)$ cost ($O(1)$ when pure random sampling without any sorting). Diverse

Table A2: Compare KID with additional baselines. Note that only FiD (Izacard & Grave, 2021b;a) is explicitly optimized for knowledge awareness. Diverse Decoding (Baheti et al., 2018) aims to improve generation diversity by constraining decoding distribution with topic and semantic similarity. PPLM (Dathathri et al., 2020) uses a static set of tokens as global constraints to do conditional generation. KID differs from all these methods as its knowledge awareness and dynamic nature. Little KID is a naive implementation of KID that uses plain list instead of trie to store knowledge.

| Method | ELI5 | | MSMARCO | | PIQA | | PubMedQA | |
|---|---|---|---|---|---|---|---|---|
| | B-1 | R-L | B-1 | R-L | B-1 | R-L | B-1 | R-L |
| FiD-T5 (base) | 15.3 | 11.3 | 45.3 | 47.3 | 12.9 | 11.9 | 4.0 | 4.4 |
| FiD-T5 (large) | _15.7_ | _18.1_ | _51.9_ | _53.1_ | _14.8_ | _17.4_ | 4.0 | 4.5 |
| Diverse Decoding | 4.6 | 15.7 | 27.5 | 22.9 | 11.3 | 15.5 | 3.5 | 3.8 |
| PPLM | 11.9 | 15.0 | 23.4 | 26.6 | 10.3 | 11.9 | 3.8 | 4.2 |
| Little KID (_ref._) | 12.1 | 14.3 | 40.2 | 46.3 | 12.2 | 13.5 | _5.2_ | _6.6_ |
| **Ours:** $KID_{GPT2}$ | **27.9** | **26.6** | **47.4** | **53.9** | **17.7** | **18.4** | **9.7** | **11.5** |

Decoding, PPLM, and KID are similar as all are inference-time optimization (not necessarily need model fine-tuning, denoted as "LM FT / –"), but KID requires additional off-line knowledge trie construction (nearly the same time cost as RAG and FiD). During inference, KID takes constant time for query (as discussed in §A.2.2), and several iterations policy gradient (by constant number of steps) to guide the generation. We also analyze the space cost of KID and other baselines (Table A4). The difference between KID and RAG / FiD is the external and local knowledge memory, which is relatively small (normally < 10Mb) from our observation. Other decoding methods do not have the ability to infuse knowledge, and thus do not bring additional space cost.

Table A3: Time complexity analysis of KID and other baselines. We study the time consumption during knowledge retrieving, model training, and inference (i.e., generation). KID has comparable time efficiency as RAG and FiD, but outperforms them in knowledge infusing (discussed in the §4.2). Note that KID can function as a standalone module operating only at inference time, rather than rely on a specific model fine-tuning or pre-training together (such as RAG and FiD).

| Method / Time | Retrieving | Training | Inference |
|---|---|---|---|
| Beam Search | – | – | $O(k \log k)$, $k = \#$ of beams |
| Sampling | – | – | $O(1)$ or $O(k \log k)$, $k = $ top-$k$ |
| RAG | $O(\#docs)$, $\#docs \approx 10$ | LM FT | $\geq O(1)$ |
| FiD-T5 (base) | $O(\#docs)$, $\#docs \approx 100$ | LM FT | $\geq O(1)$ |
| FiD-T5 (large) | $O(\#docs)$, $\#docs \approx 100$ | LM FT | $\geq O(1)$ |
| Diverse Decoding | – | LM FT / – | Two Neural Models Prediction + $O(1)$ |
| PPLM | – | LM FT / – | $O(\#\text{control words} * \text{step})$ |
| **Ours:** KID | $O(\#docs) + DFS$, $\#docs \approx 5$ | LM FT / – | $O(h_{\max}^2) \approx O(1)$ |

## A.4 SAMPLE GENERATIONS

We list several generation samples of KID and RAG. In general, the generated texts of both methods are highly readable; however, we find RAG's generation tends to off-topic and not coherent with current on-going concepts flow when the generation is long (e.g., ELI5, "_smoking inside the vehicle_"), or the context is multi-fold and complicated (e.g., multi-turn dialogue WoW, "_do some exercise_")— these are legit continuation if we only consider immediate context. KID is superior especially in longer generation, which seems to echo our findings that KID can mitigate exposure bias.

We show some generation samples of KID and RAG in Table A5.

## A.5 ACTUAL QUESTIONS WE ASK HUMAN ANNOTATORS

We asked participants about: 1) **Relevance** (i.e., _"What's the relevance level between the generated text and the given context?"_ Answer is from 1-_totally not relevant_ to 7-_totally relevant_), 2) **Factuality**

Table A4: Memory footprint of KID and other baselines. Similar to RAG and FiD, KID requires pre-store the grounding documents on disk (and corresponding dense vectors). In addition, KID builds a knowledge trie off-line (less than 10Mb for all the tasks we studied), and a local knowledge memory (a limited length FIFO list), to enable knowledge infusion during generation.

| Method / Space | External Resources / Model | Knowledge? |
|---|---|---|
| Beam Search | – | ✗ |
| Sampling | – | ✗ |
| RAG | 13.4G Docs + 64.6G Index / BART | ✓ |
| FiD-T5 (base) | 13.4G Docs + 64.6G Index / T5 | ✓ |
| FiD-T5 (large) | 13.4G Docs + 64.6G Index / T5 | ✓ |
| Diverse Decoding | Two Models 2.8G in Total / LM | ✗ |
| PPLM | BoW or Classifier / LM | ✗ |
| **Ours:** KID | 13.4G Docs + 64.6G Index + $G_{loc}$ + $G_{ext}$ / LM | ✓ |

Table A5: Sample generation of KID and RAG on ELI5 (QA), ROC (Logic-centric Writing), and WoW (Dialogue) with fine-tuned language models.

| | ELI5 |
|---|---|
| Question | Does marijuana impair driving ability? |
| RAG | Marijuana does impair your ability to drive, but you can avoid this by not being asleep together in class together and using a napkin for balance.. One way to avoid discomfort would be smoking inside the vehicle. That being said, smoking in public can do the same thing as smoking outside. |
| KID | Marijuana does impair your ability to drive because cannabis is a depressant drug, which means it slows reaction times and information processing. A person who has been using cannabis is more likely to cause a vehicle collision, but may still think that they can drive safely. |
| | **ROC** |
| Context | Keith lived in an area that had occasional earthquakes. But he thought keeping an first aid kit was unnecessary. The next day, he cut his hand while cutting food. And he had to drive himself to a hospital. |
| RAG | Keith quickly became worried and sad. |
| KID | Keith eventually learned first Aid at a clinic instead of building his own kit. |
| | **WoW** |
| Utterance | f : would you like some tea or coffee ? m : no thank you . it 's very late now , that will keep me awake the whole night . f : then what about some water ? m : yes , please . f : do n't work too late since you were not in good health . you should be careful with your health . |
| RAG | m : thank you. i would sleep earlier and do some exercise in the morning |
| KID | m : thank you for your concern. could i have some water please . |

(i.e., *"What's the factuality level of the generated text?"* Answer is from 1-*totally not factual* to 7-*very well factual*), and 3) **Grammaticality** (i.e., *"How much the text is similar to human-generated text?"* Answer is from 1-*not similar at all* to 7-*very much similar*).

