# OpenReview forum: "Knowledge Infused Decoding"
_ICLR.cc/2022/Conference — ICLR 2022 Poster_

### Official Review · Reviewer_Kmpi · 2021-11-01

**Correctness:** 3
**Technical Novelty And Significance:** 3
**Empirical Novelty And Significance:** 2
**Recommendation:** 6
**Confidence:** 4

**Main Review:**

Strengths:
This approach is interesting and very relevant to the growing work on retrieval- and memory-augmented models for grounded generation.
Since it is a new decoding approach, it is model and task agnostic, and can be used as a plug-in replacement for existing decoding approaches without changing the architecture and for grounding each generated token in external knowledge. The knowledge graph could be further used to study provenance of generated tokens.
Experiments are conducted on a variety of knowledge-intensive tasks/datasets and KID clearly does outperform vanilla beam search and sampling which are not grounded.

Weaknesses:
There is a lack of clarity and details which makes it hard to follow some explanations and results; choice of baselines seems a bit unclear -- digging into these points through the following questions to allow for discussion.
1. Since probability mass is shifted in favor of retrieved demonstrations at every time step, how is the model generating words that don’t appear in those demonstrations? Eg: stop words for fluency. What is happening in those cases?
2. During constrained decoding, are the concepts always the subjects (keys) in the triples? If so, then is the local trie reconstructing portions of the external trie? If not, how is the local trie used to traverse the external trie?
3. What is the “in-house split dev set”? No official test sets are used, right? The number for RAG on MS MARCO in Table 2 (40.8 R-L) appears to match the one from the original paper for test scores (https://arxiv.org/abs/2005.11401 their table 2). But the dev set scores in their table 6 seem to be higher (57.2 R-L). It’s a bit confusing. Could you please clarify which subsets were used for each of the evaluations and what is the fair comparison?
4. Are the beam and sampling baselines in Table 1 conditioning on external knowledge? If not, then aren’t the baselines far too weak for a fair comparison and should include some external knowledge? The published SotA numbers appear to paint a very different picture and that seems to call the claims into question. On a related note, why not also compare against other constrained decoding baselines eg: lexically constrained decoding (https://arxiv.org/abs/2010.12723)?
5. Could you please elaborate on the runtime details? Providing a breakdown for the different steps might even be useful. How does this compare to vanilla decoding methods? How does it compare to RAG-token/kNN-LM/other related approaches?
6. Fusion-in-Decoder (https://arxiv.org/abs/2007.01282v2) seems like a very important baseline for knowledge infusion. Was there a reason to not include it?


Some additional comments:
1. RAG-token (https://arxiv.org/abs/2005.11401) supports token-level retrieval, so does kNN-LM (https://arxiv.org/abs/1911.00172). Discussion in the intro and some other places appears to misrepresent the former and leave out the latter. It might also be worth highlighting that RAG-Sequence is used for all the comparisons.
2. Would it be possible to add a table somewhere (perhaps the appendix) that provides the dataset sizes and other relevant stats?
3. Would it be possible to add some qualitative examples? It would be useful to see, for instance, what a single timestep’s retrieved demonstrations, local trie etc. look like.

=============================

Update: Thanks to the authors for their detailed response. While the responses helped to clarify a lot of details, there remain some issues particularly regarding the clarity of the paper itself. That being said, the community stands to benefit from ideas presented. Recommending acceptance.

Some suggestions for revisions:
- Not convinced that claiming the win over beam search/sampling is a notable contribution since a large body of work has already shown that injecting knowledge is useful. Would suggest the authors consider reframing this -- it’s fine to include the experimental results, but the argument and claims need to at least be thought out.
- Probably update the text to reflect what changes have been made to MS Marco results so it’s clear what the evaluation procedure is.
- Adding all the response material to the appendix doesn’t help the clarity of the paper itself, might help to coherently revise sections.
- Need to clarify the discussion on runtime. It’s a bit tedious to parse.


**Summary Of The Paper:**

This paper presents Knowledge Infused Decoding (KID), an RL-based approach for grounded decoding by conditioning on external knowledge. For every new example:
- retrieve the relevant passages from Wikipedia (using dense passage retrieval DPR)
- construct a knowledge trie of triples extracted from the passages and initialize a local trie that tracks all the nouns/verbs mentioned
- Generate tokens conditioned on both prior context and by retrieving triples from the knowledge trie, used as demonstrations for policy gradient
This approach is model-agnostic and can be used to fine-tune both encoder-decoder and decoder-only models.

Experiments are divided into two parts:
- evaluating the decoding method, on six NLG datasets -- outperforming beam search and (top-k/nucleus) sampling
- evaluating the knowledge infusion, on four abstractive QA datasets -- outperforming a number of approaches including retrieval-augmented, graph neural network, and knowledge graph methods.

Ablation studies are presented for swapping out the retriever, different hyperparameters for KID and the size of the underlying LM.


**Summary Of The Review:**

Updated to weak accept after author response.

Recommending a weak reject (5) pending the discussion on concerns raised in the main review (questions 3,4,5,6) regarding baselines and results. These seem important enough to clarify before updating the recommendation.

---

> ### Author Response · Authors · 2021-11-16
> **Response to Reviewer Kmpi**
>
> Thanks for reviewing our paper, and providing your valuable feedback! We are glad that you found our idea interesting, experiments extensive and convincing. We have revised our paper to address your concerns (Please take a look at our general response for highlights of the revision). Below we answer your questions:
>
> **About Weaknesses:**
> 1. **Stop words?** We’ve added detailed description about how we construct, query, and use the queried knowledge to guide generation in section A.2 of appendix of our revised paper. Typically, the query words which we store in the local memory and the keys in the knowledge trie are non-stop words, since querying stop words in the knowledge trie cannot lead to meaningful constraints. As described in A.1, the local memory only appends non-stop-word entities, so if the current generated token is a stop word, we will skip it (not appended to the local memory, and thus no query happens). Note that in Equation 3 the knowledge-constrained policy is actually derived from vanilla policy with a trust region limited by KL divergence---the guidance is thus a soft constraint and the vanilla policy will take over the generation if no query for knowledge is needed (to keep the generation fluent and coherent).
>
> 2. **Local Memory.** The local knowledge memory is simply a First-in-First-out entity list that grows when more tokens are generated. More details can be found in our newly added section A.1 in the appendix. We’ve also provided the pseudocode of the KID generation procedure (Algorithm 2).
>
> 3. **In-house dev set?** Yes, since some authors of the datasets did not reply back to us about the test set results in time, we followed the evaluation protocol used by Zhou et al. [1] (see Evaluation Metrics in page 6), where the hyperparameters are tuned on the in-house split dev set and the final evaluation is conducted on the official dev set. About RAG’s result on MSMARCO, we find the discrepancy is caused by that the official evaluation script uses max ROUGE-L score across the multiple references (see the [code](https://github.com/facebookresearch/KILT/blob/main/kilt/eval_downstream.py), line 150), while our implementation uses the average. The alignment to the official version leads to 6-8 points absolute improvement on ROUGE-L of KID and other baselines. We have fixed all the related results, and recorded RAG’s official dev set performance in Table 2.
>
> 4. **More baselines?** Table 1 aims to demonstrate that a normal fine-tuned model armed with a carefully designed decoding algorithm (such as our proposed KID) can achieve comparable performance to task-specific SotA models. Also, as mentioned in the paper (footnote 5), we conducted extensive experiments on finding the best hyperparameters for beam search and sampling decoding. For example, each sampling decoding result in Table 1 is the best over nearly 100 runs of different top-$p$ and top-$k$ combinations – with this extensive grid search, we aimed to reach the upper bound of the original decoding methods’ performance so that we can make sure KID makes solid improvement over vanilla decoding rather than compare it with underestimated baselines. In Table 2, we then compare with many strong knowledge-aware baselines and KID shows superior performance. We also include comparisons with more baseline in section A.3 of appendix in our revised paper. We have tried to include the constrained decoding in our additional comparison but we cannot find its official implementation (it’s still in preprint condition). We have now cited this work in our paper.
>
> 5. **Runtime?** We’ve included a dedicated section (section A.3.2) in the appendix to compare different decoding strategies (including vanilla ones like beam search and sampling) in terms of time and space consumption. As you suggested, we break down the whole process into retrieving, training, and inference stages, and analyze them separately. We compare KID with many baselines in terms of knowledge-aware performance (Table A2), time complexity (Table A3), and memory footprint (Table A4). Please take a look and let us know if any improvements can be made! Thank you!
>
> 6. **Adding FiD.** We have included comparison with FiD-T5 (base and large) in the Table A2, A3, A4 of our revised appendix. We test it on four QA tasks (two in-domain ELI5/MSMARCO, two out-of-domain PIQA/PubMedQA), since currently FiD has been only fine-tuned on QA tasks.
>
>
> **Additional Comments**
>
> We have cited the mentioned work, added Table A1 in the appendix to describe the data statistics, and presented a real example of KID knowledge construction and query in Figure A1. Please take a look at our visualization and related discussion.
>
> Finally, we thank the reviewer once again for their thoughtful comments and constructive suggestions! We hope our revised version can solve your concerns and questions!
>
> [1]: [Pre-training Text-to-Text Transformers for Concept-centric Common Sense](https://openreview.net/pdf?id=3k20LAiHYL2)

---

> ### Author Response · Authors · 2021-11-30
> **Thank you and further response**
>
> We thank the reviewer’s constructive suggestions about clarity, and we feel grateful you recommended acceptance for our work!
>
> Unfortunately we cannot upload a new version right now, but we will definitely incorporate those details into our final revision following your suggestions. Below we briefly answer your questions:
>
> **About including beam search/sampling as baselines.** We consider the most noticeable contribution of KID to be that it outperforms other knowledge-infusing baselines in few-shot cases (22% on average), and shows relatively strong results in full-shot settings (as shown in Table 2). Note that we choose four diverse QA tasks since most of these task-specific baselines are optimized for QA, while KID can serve as a task-agnostic decoding method. In Table 1 we compare KID with beam search/sampling for completeness (they are task-agnostic decoding methods as well, so they can be adapted to diverse NLG tasks). Interestingly, we find that the fine-tuned LMs armed with KID can even obtain competitive performance with respect to task-specific SotA methods (in Table 1). In our revised version we will reorganize these sections and highlight our main contributions as suggested by the reviewer.
>
> **About other suggestions about clarity.** Thanks for your comments! We will reflect our additions to the appendix in the main body of paper.
>
> Hope this further addresses your concerns. Thank you!

---

### Official Review · Reviewer_zdRT · 2021-11-02

**Correctness:** 3
**Technical Novelty And Significance:** 3
**Empirical Novelty And Significance:** 2
**Recommendation:** 5
**Confidence:** 2

**Main Review:**

## Strengths

- Proposed method outperforms baselines on various knowledge-intensive tasks.
- Proposed objective can be a good loss term for explicitly modeling knowledge in LM.
- Considering the performance gap between KID and RAG (or BART+DPR), dynamic knowledge infusing will be a good alternative to static knowledge grounding.

## Weaknesses

- Knowledge-graph construction and graph traversal are not well explained. Can you give an example knowledge trie and explain to me how does the graph traversal work? How would the queried results look like?
- Optimization looks somewhat tricky. Which learning rate is used for Adam? How did you select K = 3 steps for policy update? What would happen if we set K < 3 or K > 3?
- Can you show me example sentences generated by KID and compare them with RAG-generated sentences?
- In Table 5, can you also measure naturalness or grammaticality? It is obvious that KID will improve knowledge grounding, but not sure how much it will break the syntax of generated sentences.
- Can you also add FiD-BART as one of the main baseline methods?

**Summary Of The Paper:**

This paper deals with the well-known but important problem of language models -- enhancing language model with (external) knowledge. To that end, the authors present Knowledge Infused Decoding (KID), a novel decoding algorithm for generative LMs, which infuses external knowledge into each step of the LM decoding. This method maintains a local knowledge memory based on the current context and continuously updates the local memory via reinforcement learning to guide each step of language generation. As a result, it can integrate knowledge on the fly and perform well on various knowledge-intensive tasks such as abstractive QA, logic-centric writing, and dialogue generation.

**Summary Of The Review:**

The proposed method is fairly sound and shows good empirical results. However, some details are missing (e.g., knowledge graph construction and traversal), and hyperparameter selection is not well explained (e.g., learning rate and the number of optimization steps). It is also unclear how much KID will harm the original LM in terms of naturalness and grammaticality. Additional human evaluation or generation examples can remedy these issues.

---

> ### Author Response · Authors · 2021-11-16
> **Response to Reviewer zdRT**
>
> Thank you for your valuable feedback. We appreciate that you found our paper’s method sound with respect to experimental procedure. Below we address the questions raised in your comments:
>
> - **Weakness 1: More Explanation about Knowledge Memory Construction and Query.** In the appendix of the revised version, we prepare a dedicated section (section A.2) to discuss a) how we construct the local memory and external knowledge trie, and b) how we query the external trie with local memory. We use a diagram (Figure A1) to visualize what the knowledge trie looks like and describe the query procedure in A.2.2. To help with the understanding, we also provide pseudocode (Algorithm 2) to show KID’s entire generation loop. We incorporate references to these newly added content in the main body of our paper as well.
>
> - **Weakness 2: Learning Details.** We use 3e-4 as the learning rate of Adam optimizer. Overall, $K$=3 works the best in terms of our empirical observation. We found that $K$ < 3 leads to small updates on the policy which means the generation approaches the normal MLE trained model behavior, while $K$ > 3 leads to dull generation since now the token distribution is heavily limited by the retrieved knowledge, which is not ideal for tasks that require longer and creative generation (e.g., long-form QA ELI5, and ROC story generation). For example, in the ELI5 task, we find the generation under $K$=4 or 5 includes several repetitions. Even higher $K$ results in incoherent sequences. In our testbed (in total 8 NLG tasks) $K$ = 3 seems to be an optimal choice for most cases given the model and optimizer.
>
> - **Weakness 3: Example Generation.** Yes, we now include some sample generations in section A.4 of the appendix of our revised version paper. Please take a look at Table A5.
>
> - **Weakness 4: More Human Evaluation.** Thanks for your suggestion. We have now added results of the human evaluation of grammaticality of responses in Table 5, and briefly discuss the results in Section 4.4.
>
> - **Weakness 5: Add FiD as one of the Baselines.** Thanks for pointing this out. We’ve added comparison with FiD in the Table A2 (performance), Table A3 (time complexity), and Table A4 (memory footprint) in the appendix of our revised paper. After digging into their code we found that only the FiD + T5 model (base and large) checkpoints are officially released. Thus, for a fair comparison, we chose the T5 version of FiD and tested the base and large models on four QA tasks (two in-domain ELI5/MSMARCO, two out-of-domain PIQA/PubMedQA), since currently FiD has been only fine-tuned on QA tasks. The detailed results and analysis can be found in the appendix. We also include FiD as one of the main baselines in the main body of the revised paper (see Table 2).
>
> Finally, we appreciate the suggestions on the paper presentation and missing baselines. We hope our revised version could solve your concerns. Thanks again for your thoughtful reviews!

---

### Official Review · Reviewer_wRJi · 2021-11-03

**Correctness:** 3
**Technical Novelty And Significance:** 3
**Empirical Novelty And Significance:** 3
**Recommendation:** 8
**Confidence:** 4

**Main Review:**

Strengths:
- This method does not require re-training or fine-tuning LMs with knowledge based objectives. This is a plug and play decoding strategy that competes with other sampling based or beam based decoding strategies with weighted decoding methods.
- This method can 'dynamically' infuse external knowledge into each step of LM decoding. This is an advantage over static knowledge retrieval based methods that retrieve documents only once and suffers from problems with long range generation.
- The probability mass of tokens at each step of token generation can be informed directly by the concepts and relations extracted via the entity-relation extraction methods; this token level reinforcement helps with overcoming sparse rewards generally popular with sequence level reward induction (via evaluation metrics such as BLEU)
- Results show that larger LMs benefits Knowledge infused decoding(KID) method and outperforms Beam search and sampling decoding. In addition, KID brings more gain for non-finetuned LMs than fine-tuned ones, potentially because the fine-tuning LM is already fit onto the new domain thus less knowledge infusion is needed.


Weaknesses:
- The paper lacks details of the text decoding inference time and comparison with other decoding methods (sampling) including knowledge based decoding methods. Figure 2(a) and 2(b) do show time latency; however there is no comparison or discussion of this in the paper.
- The paper presents a text decoding method for knowledge infusion; however, it does not compare this knowledge infused decoding with other non-RL based decoding techniques such as penalized sampling of words (Keskar et al. 2019), decoding with distributional constraints (https://arxiv.org/pdf/1809.01215.pdf) or other plug and play decoding methods such as PPML (https://arxiv.org/abs/1912.02164).
- The ablation studies described in the paper do not make explicit the contribution of the RL based decoding technique.
- The trie data structure is created using open IE tools and reference previous literature. However, some more analysis about the compression of memory would be interesting to understand (with availability of longer sequence handling transformers, how important is the role of compression and trie based external memory?)
- It is not clear how dynamic is the local memory; Figure 1 is not well-described in the paper. At time-step t or t+1, it seems that the local memory is not going to change (extracted from context); what are the dynamics of local memory in this model?



**Summary Of The Paper:**

This paper presents a text decoding strategy for knowledge-intensive text generation tasks with the goal to infuse external knowledge based information at each time-step of decoding process via efficient document retrieval, creation of local and fast global knowledge access data structure and the interaction guided decoding method. The decoding method uses RL based policy gradient for token selection. The method is evaluated using diverse types of knowledge-intensive tasks such as abstractive question answering, logic centric writing and dialog generation. This method works with existing decoder only or encoder-decoder LMs as a plug and play method and outperforms existing knowledge aware task optimized models and correlates well with human evaluation.

**Summary Of The Review:**

This paper presents a novel plug and play text decoding method that allows improved text generation for knowledge intensive tasks via reinforcement learning based technique. The method allows several new future investigations in this direction and is very interesting (addressing exposure bias problems with LM tuning). There's work on knowledge retrieval, data structures for knowledge encoding and text decoding with state of the art integrated methods. The results are also superior compared to many existing decoding and knowledge aware decoding methods.

---

> ### Author Response · Authors · 2021-11-16
> **Response to Reviewer wRJi**
>
> We thank the reviewer’s thoughtful comments and suggestions, and we are glad that you found our idea interesting and novel. Below we answer your questions:
>
> - **Weakness 1 & 2: Comparison with other Decoding Methods, and Time and Space Analysis.** In section A.3 of the updated appendix, we have compared KID with several more baselines. Section A.3.1 includes comparison with Diverse Decoding, PPLM, and CTRL (cannot be directly compared but we discuss its difference from KID). In section A.3.2, we analyze and compare the time and space consumption of KID with other decoding-related methods, including vanilla beam search, sampling decoding, and some other knowledge-aware methods. Please let us know if any improvements can be made! Thank you!
>
> - **Weakness 3: Ablation to study the RL Contribution.** In Table 1, besides the fine-tuned (FT) LMs armed with KID, we also show the performance of fine-tuned LMs armed with sampling and beam search decoding, aiming to show the effectiveness of RL-guided KID. Note that the fine-tuning procedure is based on the MLE objective so the results demonstrate that a pure MLE stage cannot reliably infuse knowledge during generation, and that KID’s MLE pre-training + RL calibrated decoding formulation is a superior solution. Similar settings are also used to study the exposure bias problem (Figure 2 (c)), where the generation quality of beam search and sampling (pure MLE) deteriorates as longer sequences are generated, while KID (MLE + RL) can maintain a more stable quality in terms of human judgement. In general, we conclude the contributions of RL are 1) more effective knowledge infusion into generation, and 2) mitigating the exposure bias problem when generating longer sequences.
>
>
> - **Weakness 4: About Data Structure Choice.** We add a naive implementation of KID (named Little KID) as a baseline and compare KID with it in section A.3.1 of the appendix. Little KID differs from KID as it does not support querying for next-step demonstrations (and by multi-hop) but uses the list of entities extracted from grounding documents as a static constraint to guide the generation. It shows inferior performance in our experiments and tends to generate generic and repeated content in longer sequences, partially because the static form of the knowledge does not align well with the dynamic nature of the generation.
>
> - **Weakness 5: About Local Knowledge Memory.** We detail how we construct external and local knowledge memory in section A.2 of the appendix. Specifically, local knowledge memory is a FIFO list with length of $w_{\textrm{max}}$. It will only append entities extracted from the newly generated tokens. Section A.2 also describes the knowledge trie querying process and how it interacts with the local memory. Please take a look and let us know if any improvements can be made!
>
> Thanks again for your thoughtful reviews and helpful suggestions!

---

### Official Review · Reviewer_RDPE · 2021-11-05

**Correctness:** 3
**Technical Novelty And Significance:** 3
**Empirical Novelty And Significance:** 3
**Recommendation:** 6
**Confidence:** 3

**Main Review:**

Strengths:
- The proposed method can be considered as a new approach for constrained decoding, but rather than doing it in a hard way, the proposed method gives a soft constraint by rewarding a set of target words.
- The experiments were done in multiple datasets and the proposed method consistently shows a better result than its baselines. It is especially encouraging because the proposed method is a plug-and-play module that can be applied to any language model in practice.

Weaknesses:
- The paper is overall difficult to understand, and I had to re-read it multiple times to make sense of it. This might have caused several misunderstandings.
- The paper does not compare against vanilla (non-graph) constrained decoding which limits the output space to the words in the retrieved corpus. This will be an important baseline that can show the advantage of having such graph for constrained decoding.

Questions:
- I am confused with the motivation behind formulating the loss with RL. The paper discusses the disadvantages of vanilla decoding, e.g. exposure bias of teacher forcing. However, the proposed loss function with reward at every time step is also biased. The proposed loss seems to me more like an MLE with some regularization/modification.

**Summary Of The Paper:**

The paper is interested in improving the performance of natural language generation tasks that require external knowledge (which are often called knowledge-intensive tasks). In particular, the paper focuses on how it can leverage existing language model (e.g. GPT-2, BART) without changing its architecture and injecting external knowledge into it. To do so, the paper proposes Knowledge Infused Decoding (KID), which alters the decoding probability at each time step (in vocab space) so that it favors decoding certain words, which are obtained by retrieving a relevant document from a corpus. In order to trains such decoder, the authors also add a KL divergence to the loss that prevents the decoding distribution not to be too different from the original Language Model's decoding distribution. When KID is used in conjunction with GPT2-M or BART-L, they show clear improvements, and in some datasets, they achieve higher accuracy than the state of the art. The paper also shows that KID outperforms Retrieval Augmented Generation (Lewis et al., 2020) which uses the retrieved documents directly into the input (whereas KID uses it indirectly to constrain the decoding in a soft way).

**Summary Of The Review:**

I think the paper's result is great and I agree with the overall motivation behind the paper. However, many parts of the paper are unclear that I will need further clarification to correctly assess this paper.

---

> ### Author Response · Authors · 2021-11-16
> **Response to Reviewer RDPE**
>
> We are glad that you found our idea novel, our results great, and the conclusion encouraging. We have revised our paper to address your concerns (Please take a look at our general response for highlights of the revision as well). Below we provide responses to your questions and concerns:
>
> - **Weakness 1: About Clarity.**
> In the newly added appendix, we have added details on how we construct the knowledge trie, and how we query it during generation. We’ve also added a diagram (Figure A1) to visualize the querying process, and the pseudocode (Algorithm 2) to show KID’s entire generation loop. We refer to this new appendix section in the main body of the paper as well. We hope these attempts can present our idea more clearly.
>
> - **Weakness 2: Comparison with Naive Implementation Version of KID.**
> Thanks for your suggestion. We had tried the vanilla solution (not using graph data structure) in our preliminary experiments. We now include the results from such a baseline (named Little KID) in section A.3 of the Appendix for comparison. It performs inferiorly compared with graph-based KID, since it cannot dynamically query for next-step demonstrations but uses a static set of entities to guide generation, which leads to noisy guidance signals and generic generation sometimes. However, naive KID shows relatively good results in tasks requiring a long generation while having a small dataset (e.g., PubMedQA, which has only 800 training samples, and 200-token references on average), which we conclude is the advantage of inference-time optimization. FiD (and RAG) infuses knowledge during training but has no scheme to guarantee that the knowledge will be expressed during generation. We’ve added references to these takeaways in the main body of the paper as well. Please let us know if you have any further improvement suggestions. Thank you!
>
> - **Question: About Formulating the Loss with RL.**
> Compared to MLE which treats each token equally (such as predicting missing tokens given nearby context in masked LMs like BERT), the formulation of RL enables us to give more importance to the concept-carrying tokens. Policy-based RL helps us directly optimize the (dynamic) policy rather than a (static) reward function, thus making it a good fit for the generation scenarios. The regularization term appears in the PPO [1] and TRPO [2] papers as well, and is aimed at providing a trust region for optimization, which we found to work well empirically as well. In general, we attempt to go beyond the original MLE object and reformulate the generation process as a RL procedure to enable fine-grained control.
>
> Thanks again for your thoughtful reviews and helpful suggestions!
>
> [1]: [Proximal Policy Optimization Algorithms](https://arxiv.org/pdf/1707.06347.pdf)
>
> [2]: [Trust Region Policy Optimization](https://arxiv.org/pdf/1502.05477.pdf)

---

### Author Response · Authors · 2021-11-16
**General Response to All Reviewers**

We thank all the reviewers for their valuable comments and constructive feedback. We have revised the paper accordingly (highlighted in bronze in the paper). We summarize the highlights from the revision below and address each reviewer’s feedback separately as well.

- **More details on how KID performs knowledge construction, and decoding (suggested by R1, R2, R3, R4).**  In the appendix of the paper, we’ve added a dedicated section (section A.2) on how KID constructs the external and local knowledge memories from retrieved documents (A.2.1), and how KID dynamically queries and infuses knowledge for decoding (A.2.2). A concrete example to demonstrate the process is also shown in Figure A.1. Pseudocode summarizing the detailed generation is included in Algorithm 2.

- **More comparison with other constrained generation baselines (suggested by R2, R3, R4).** We’ve added experiments comparing KID with Diverse Decoding (R2), PPLM (R2), and FiD-T5 (R3, R4) in section A.3.1 of the appendix. Overall, KID consistently outperforms these added baselines, demonstrating its superior ability to infuse knowledge for knowledge-intensive generation tasks. Some of these baselines are now included in the main body of the revised paper as well.

- **Time and space complexity analysis (suggested by R2, R4).** We’ve added a dedicated section in the appendix (section A.3.2) to decompose and analyze the time and space complexities of KID, and compare it with the baselines. Overall, KID is a more efficient and effective solution for inference-time knowledge infusion, with minimal additional parameters (for external knowledge trie and local knowledge memory).

- **Generation samples and additional human evaluation (suggested by R3).** We’ve added generation samples from KID and RAG on ELI5, ROC and WoW in section A.4. Qualitatively, generations from KID are more coherent and to the point for the provided context. We’ve also included results from additional human evaluation of grammaticality of generation in Table 5 of the main body of the paper.

Finally, we are glad that reviewers found strengths in our paper’s novelty, experimental procedure, and contribution to the community. Different from existing methods, KID aims to serve as a model-agnostic inference-time solution to infuse knowledge into language generation. We hope our revised paper can clarify the confusing parts in our previous version.

---

### Decision · Program_Chairs · 2022-01-20

**Decision:**

Accept (Poster)

**Comment:**

The paper introduces a novel decoding algorithm that allows to dynamically integrate external knowledge with generative
LMs. The proposed technique is plug-and-play, it does not require re-training or fine-tuning LMs with knowledge based objectives.
The author report a series of experiments on several datasets and tasks showing improvements over competitive baselines and in some
cases above SOTA. The authors have addressed the reviewers' queries and added experimental results (additional baselines) as
well as clarifications of their approach (e.g., Figure A1 in the Appendix). I think the topic  (e.g., constrained LM decoding) is of general interest to the ICLR community and the approach compelling.